# An Accessible Diagnostic Toolbox to Detect Bacterial Causes of Ovine and Caprine Abortion

**DOI:** 10.3390/pathogens10091147

**Published:** 2021-09-06

**Authors:** René van den Brom, Inge Santman-Berends, Remco Dijkman, Piet Vellema, Reinie Dijkman, Erik van Engelen

**Affiliations:** 1Department of Small Ruminant Health, Royal GD, P.O. Box 9, 7400 AA Deventer, The Netherlands; p.vellema@gdanimalhealth.com (P.V.); r.dijkman2@gdanimalhealth.com (R.D.); 2Department of Research and Development, Royal GD, P.O. Box 9, 7400 AA Deventer, The Netherlands; i.santman@gdanimalhealth.com (I.S.-B.); r.dijkman@gdanimalhealth.com (R.D.); e.v.engelen@gdanimalhealth.com (E.v.E.)

**Keywords:** ovine, caprine, abortion, bacterial agents, accessible diagnostic toolbox, fetal oropharyngeal mucus sample, fetal lung puncture sample, sheep, goats

## Abstract

Results of laboratory investigations of ovine and caprine cases of abortion in the lambing season 2015–2016 were analyzed, using pathology records of submissions to Royal GD (Deventer, the Netherlands) from January until and including April 2016, in comparison with the results of two accessible alternative techniques for sampling aborted lambs and kids, swabbing the fetal oropharynx and puncture of the fetal lung. *Chlamydia abortus* was the main cause of abortion in sheep as well as in goats. Other causes of abortion were *Campylobacter* spp., *Listeria* spp., *Escherichia coli*, and *Yersinia enterocolitica.* Ovine pathological submissions resulted more often in detecting an infectious agent compared to caprine submissions. For the three main bacterial causes of abortion, *Campylobacter* spp., *Listeria* spp., and *Chlamydia* spp., compared to results of the pathological examination, oropharynx mucus, and fetal lung puncture samples showed an observed agreement of 0.87 and 0.89, an expected agreement of 0.579 and 0.584, and a kappa value of 0.691 and 0.737 (95% CI: 0.561–0.82 and 0.614–0.859), respectively. The agreement between the results of the pathological examination and both fetal lung puncture and oropharynx mucus samples was classified as good. In conclusion, although a full step-wise post-mortem examination remains the most proper way of investigating small ruminant abortions, the easily accessible, low-threshold tools for practitioners and farmers as described in this paper not only provide reliable results compared to results of the post-mortem examination but also stimulates farmers and veterinarians to submit fetuses and placentas if necessary. Suggestions for further improvement of both alternatives have been summarized. Both alternatives could also be tailor-made for specific regions with their specific causes of abortion.

## 1. Introduction

Abortion, the expulsion of a fetus that is incapable of independent life, premature birth, the expulsion before full term of a fetus capable of independent life, and stillbirth, the expulsion of a dead full-term fetus, together are here referred to as abortion [1]. In small ruminants, abortion can be caused by various non-infectious and infectious agents, and several of these infectious agents constitute a zoonotic risk [2]. As a consequence, abortion not only results in reproductive losses and economic damage, and concerns about the possible disease for the owner, his family, farm visitors, and, when *Coxiella burnetii* is involved, people living in the surroundings [3,4]. 

Monitoring of the causes of abortion [5] is an important part of the approach to contain these risks, and in the Netherlands, Royal GD (GD) is mandated by her stakeholders to perform post-mortem examinations on submissions of ovine and caprine fetuses and placentas. However, decreasing submissions are a source of concern as to whether the results reflect reality [5].

For laboratory investigation of ruminant abortion, a step-wise investigative approach was recommended by Borel et al. (2014) [6]. This approach consists of (1) registration of the case history, (2) sampling of maternal blood for serology, (3) performing a macroscopic examination of the fetus and placental membranes, (4) sampling of fetus and placenta for microbiology, histopathology, and molecular analysis, (5) performing routine bacteriology, (6) and histopathological examination, (7) using immunohistochemistry or *in situ* hybridization to demonstrate the presence of pathogens, and (8) immediate reporting of notifiable diseases. Under optimal circumstances, this approach should be performed for each submission of ovine and caprine abortion. However, on many occasions, several factors hamper this approach, like financial limitations, doubt whether all those efforts lead to an aetiological diagnosis, transport of fetuses and placentas, and availability of fully equipped laboratories within an acceptable distance. These limitations have stimulated the development of alternatives that are easier for sheep and goat farmers and their first-line veterinary practitioners.

In the Netherlands, bacteria are the most common cause of cases of small ruminant abortion submitted to GD [5,7]. 

Based on the above, GD explored the feasibility of two alternative sampling systems to investigate bacterial causes of ovine and caprine abortions: a swab taken from the oropharynx, and a puncture sample from the fetal lung of the aborted fetuses. Test results of these samples were compared with each other and with results from standard post-mortem examinations. In this paper, we describe these sampling systems, test results, and observations on causes of abortion in small ruminants in the Netherlands during the lambing season 2015–2016.

## 2. Results

### 2.1. General Results

During the lambing season of 2015–2016, 76 submissions of ovine (n = 48) and caprine (n = 28) abortion were presented for post-mortem examination to GD. These submissions originated from 68 unique sheep (n = 45) and goat farms (n = 23). In total, 153 fetuses were submitted. A post-mortem examination revealed a farm-level diagnosis on eight out of 23 unique goat farms (35%) and 33 out of 45 unique sheep farms (73%). Out of the caprine submissions, 16 out of 28 were complete (57%). Out of the ovine submissions, 33 out of 48 (69%) were complete. The chance of making a diagnosis at post-mortem examination was significantly higher (*p* = 0.003) when the submission was complete and consisted of the fetus and placental membranes. In total, from 146 (83 ovine and 63 caprines) out of 153 fetuses that were submitted for post-mortem examination, samples from the oropharynx and/or fetal lung were collected. It was not possible to collect samples from seven fetuses that were mummified (Table 1).

### 2.2. Results for the Different Agents That Were Detected

#### 2.2.1. *Chlamydia* spp.

*Chlamydia* species were demonstrated by immunohistochemistry on the placental membranes from nine fetuses, five of ovine and four of caprine origin, in seven submissions from seven different farms. In eight out of nine fetal samples from the oropharynx mucus and the lung puncture, PCRs were positive for *Chlamydia abortus/psittaci*. In the remaining fetus, another *Chlamydia* spp. was found, and only in the lung puncture sample. 

In oropharynx mucus and lung puncture samples from fifteen fetuses, *Chlamydia* spp. was demonstrated by PCR, but not at post-mortem examination. If pathology is strictly followed as a gold standard, these results should be considered as false-positive results. These fifteen fetuses originated from ten submissions from eight unique farms. One submission was incomplete, and *Toxoplasma gondii* was considered as a cause of the abortion based on pathological examination. In the remaining submissions, a diagnosis on submission level was abortion by *Chlamydia* spp. on six occasions, based on findings in other fetuses within the submission. 

In four fetuses, *Chlamydia* spp. (*C. abortus* or *C. psittaci* once, and other *Chlamydia* species three times) were demonstrated in mucus samples from the oropharynx, while the lung puncture samples and pathology results of the same fetus were negative. These were all caprine fetuses and submitted by three different farms in three different submissions. In all these submissions, no diagnosis was made by pathological examination, and only one of these submissions was complete and contained placental membranes. In seven fetuses, *Chlamydia* spp. (*C. abortus* or *C. psittaci* once, and other *Chlamydia* spp. six times) were demonstrated in the lung puncture sample, while the mucus sample from the oropharynx and pathological examination of the same fetus were negative for *Chlamydia* spp. These fetuses originated from six unique farms in seven submissions, of which five were from ovine and two from caprine origin. In four of these submissions, no diagnosis was made at the pathological examination; in one submission, placentitis was found, in another toxoplasmosis was confirmed, and in one ovine submission *Listeria ivanovii* and *Bacillus licheniformis* were found to be the cause of the abortion. 

#### 2.2.2. *Coxiella burnetii*

In none of the 76 submissions, *C. burnetii* was demonstrated as a cause of abortion at pathological examination. Two oropharynx mucus and lung puncture samples were *C. burnetii* PCR low positive (<1000 bacteria/mL), and three samples from the oropharynx originating from one submission were *C. burnetii* PCR positive, while no placentitis was found at the pathological examination. Lung puncture samples were *C. burnetii* PCR negative. 

#### 2.2.3. *Campylobacter* spp.

*Campylobacter* spp. that were found were *Campylobacter fetus*, *Campylobacter jejuni*, * Campylobacter hyointestinalis*, *Campylobacter coli*, and *Campylobacter sputorum*. In mucus samples from the oropharynx, *Campylobacter* spp. were found eleven times, of which nine were in ovine fetuses and two in caprine fetuses. In fetal lung puncture samples, *Campylobacter* spp. was found in seven ovine fetuses. At the pathological examination, *C. fetus.* was found in four ovine fetuses, and *C. sputorum* in two ovine fetuses. In total, *Campylobacter* spp. was found in 13 fetuses from ten submissions from ten farms. In five ovine fetuses, from three submissions, from three farms, *Campylobacter* spp. were found in oropharynx mucus and fetal lung puncture samples and during pathological examination. 

#### 2.2.4. *Listeria* spp. 

In oropharynx mucus samples from six fetuses, three of ovine and three of caprine origin, *Listeria* spp. were detected. The three caprine fetuses originated from one submission. In these caprine oropharynx mucus samples, *Bacillus licheniformis* was also found. The ovine oropharynx mucus samples originated from two farms in two submissions. In all *Listeria* spp. positive cases, submissions were complete with placental membranes. In all fetuses in these submissions, *Listeria* spp. were found at post-mortem examination. A post-mortem examination of the caprine submission, *B. licheniformis* was found in one of three fetuses. In a submission of one ovine fetus, *Listeria monocytogenes* was found at the pathological examination and in the fetal lung puncture sample, while the oropharynx mucus sample was negative. 

#### 2.2.5. *Yersinia enterocolitica*


In an ovine submission consisting of two fetuses, *Yersinia enterocolitica* was found in all the samples investigated. 

#### 2.2.6. *Salmonella* spp. 

*Salmonella* spp. were not detected in any of the investigated samples. 

#### 2.2.7. Other Findings

Incidentally, agents like *E. coli*, *Streptococcus dysgalactiae* subsp. *dysgalactiae* and other *Streptococcus* spp. were found at the bacteriological examination. In 56 fetuses, no causal agent was found at pathological examination. In 22 of these fetuses, signs of inflammation were found in the placental membranes (19) or in the fetus (3). 

### 2.3. Statistical Analyses

For the three main bacterial causes of abortion, *Campylobacter* spp., *Listeria* spp., and *Chlamydia* spp., compared to results of the pathological examination, oropharynx mucus and fetal lung puncture samples showed an observed agreement of 0.87 and 0.89, an expected agreement of 0.579 and 0.584, and a kappa value of 0.691 and 0.737 (95% CI: 0.561–0.82 and 0.614–0.859), respectively (Table 2 and Table 3). The agreement between the results of the pathological examination and both fetal lung puncture and oropharynx mucus samples was classified as good. 

The sensitivity and specificity of the two alternative ways of sampling compared to pathological examination were determined individually for each of the causes of abortion and jointly (Table 4 and Table 5). 

The overall specificity of the oropharynx mucus sampling was 97.6% (Table 4). Specificity for the different agents varied between 89.9% and 100%. The overall specificity for fetal lung puncture sampling was 97.3% (Table 5). Specificity for the different agents varied between 91.4% and 100%. Results for *Salmonella* spp. were not regarded in the overall calculation. In case this would have been done, the specificity of the oropharynx mucus sampling and fetal lung puncture sampling would have been 97.8% and 98.2%, respectively.

The overall sensitivity of the oropharynx mucus sampling was 90.6% (Table 4). Sensitivity for the different agents varied between 75.0% and 100%. The overall sensitivity of the fetal lung puncture sampling was 87.1% (Table 5). Sensitivity for the different agents also varied from 75.0% to 100%.

## 3. Discussion

This study estimated the performance of two easily accessible alternative sampling methods as a diagnostic tool for small ruminants abortions. We compared these methods with the gold standard: necropsy. The samples used for this comparison were collected at the regular annual monitoring of small ruminant abortions during the lambing period 2015–2016 in the Netherlands. Several causative agents were found.

*Chlamydia abortus* was the main cause of abortion in sheep and goats, as is the case in many other European studies [8,9,10,11]. In addition, other *Chlamydia* spp. were found, although in these cases, histological signs of inflammation were not found, and therefore the role of these agents in the etiology of abortions is unclear. Nevertheless, *C. pecorum* has incidentally been described as a possible cause of small ruminant abortion [12,13]. Other major causes of abortions were *Campylobacter* spp., *Listeria* spp., *Toxoplasma gondii*, *Escherichia coli*, and *Yersinia enterocolitica.* The remaining results were in line with previous findings on abortion submissions in the Netherlands. However, *C. burnetii* has not been found as a cause of abortion in small ruminants since 2009, the starting point of the compulsory vaccination campaign for dairy goat and dairy sheep farms, farms with a public function, and sheep and goats that go to sales and shows [3,5]. Ovine submissions resulted more often in a diagnosis than caprine submissions, possibly because placental membranes were more often lacking in the latter. Complete submissions containing both fetus and placental membranes resulted in a higher percentage of diagnoses at post-mortem examination. In studies of infectious causes of small ruminant abortion, the percentage of diagnoses normally reaches 50 percent, but seldomly 75 percent. In caprine submissions, this percentage is often even lower. For successful abortion diagnosis in ruminants, input is required from producer, practitioner, and diagnostician [14]. Bacteria cause most small ruminant abortions, and a smaller percentage is of protozoal, viral, or mycotic origin [5,7,9,15,16,17]. Despite all available laboratory tests, submissions of small ruminant abortion will not always result in a diagnosis, partly because abortion can have a non-infectious cause which is not always found at post-mortem examination, but also because under some circumstances, the infectious event takes place weeks or even months before abortion occurs and the causative agent is not detectable anymore [6]. 

Small ruminant abortions are an important cause of economic loss for sheep and goat farmers. This is, however, not the only reason for farmers to investigate possible causes. Another important reason is that farmers and veterinarians increasingly realize the potential zoonotic risks of many of the infectious causes of small ruminant abortion. However, in the last decade, several countries have reported a decrease in the numbers of ovine and caprine submissions for post-mortem examination. In the Netherlands, not only practical and financial reasons make farmers and veterinarians hesitate to submit, but also the fear of the outcome of the investigations, as an increased number of abortions, is notifiable in the Netherlands, and many farmers and veterinarians do remember the culling of 55,000 pregnant goats in 2009–2010 during the Q fever outbreak [3]. 

We have investigated new accessible diagnostic tools for ovine and caprine abortions for which the samples can be collected by farmers or practitioners on the farm and shipped, conform legislation, for laboratory investigation. Both a mucus sample from the oropharynx and a fetal lung puncture sample from an aborted fetus was demonstrated to be alternatives to indicate the major bacterial causes of ovine and caprine abortion. The relative sensitivity and specificity of these alternatives, ideally 100 percent, did not reach the same level as pathological examination. One of the reasons is the way in which we analyzed the data. To be able to compare the results, we had to uniformize the findings of the different methods. This means that a positive result of pathological examination of submission was scored as the diagnosis for each separate fetus. As a consequence, the sensitivity of pathological examination increased compared with both alternatives. Results of both alternatives could also have been analyzed on the submission level, which would have increased sensitivity. However, we have chosen not to do so since this would not have been the most likely approach for sampling in the field. Under this assumption, the average sensitivity and specificity still are above 80% and 90%, respectively, and in most cases, bacterial causes were found with all three methods. 

The kappa values for the comparison of the results of fetal oropharynx mucus samples and fetal lung puncture samples with pathology results were 69% and 74%, respectively. These values are considered sufficient, given that different agents were demonstrated, different tests were used, and tests were performed on different matrices or different tissues from aborted fetuses and placental membranes. 

We showed the usefulness of the swab or puncture technique in the diagnosis of several abortifacient agents. To have an easily accessible sampling method, we only sampled the pharynx by swab or the lungs by a puncture. This means that regardless of the analysis method that is used, non-bacterial pathogens can be missed when they are only present at other locations of the fetus or fetal membranes. The use of the alternative methods in the field instead of necropsy has the consequence that another important cause of small ruminant abortion, *Toxoplasma gondii,* and a probably less important protozoal cause, *Neospora caninum*, will not be detected, since the diagnosis toxoplasmosis cannot be confirmed in both alternative sampling methods. Ideally, these two parasitic causes of small ruminant abortion require histological examination for making a diagnosis. PCR is a recommended technique to differentiate *T. gondii* from *N. caninum* associated abortions, especially when histology is not possible to be carried out [18]. The use of the alternative methods also implies that abortifacient viruses as border disease and Rift Valley fever virus will not be found. Moreover, bacteria that are associated with small ruminant abortions but are difficult to culture, like *Leptospira* spp., *Francisella tularensis*, *C. burnetii*, *Anaplasma phagocytophilum* and *C. abortus* [19,20,21] could be missed. This is partly overcome by performing a *Chlamydia* spp. PCR. The same could have been done for *C. burnetii*. Still, we decided not to do so because in another monitoring system (monthly performed bulk tank milk *C. burnetii* PCR) all dairy sheep and dairy goat farms are negative for this agent since 2016. Besides missing some infectious agents, another disadvantage of the alternative sampling methods is that they do not detect changes in tissue that indicate the causality between the detected agent and abortion. In cases where inflammation but no infectious agent has been demonstrated in the fetus or placental membranes, an infectious cause is likely [22]. Therefore, in cases where only indications of inflammation are found, farmers and veterinarians are encouraged to resubmit abortion materials to increase the chance of detecting a causal agent the next time. Moreover, in case of non-infectious causes like mineral deficiencies or teratogenic agents like bluetongue virus and schmallenberg virus, preferably fetus and placental membranes should be submitted for post-mortem examination. In cases of abortions where fetuses and placental membranes are lacking, serology of the doe or ewe might have some diagnostic value. However, positive serology for ubiquitous pathogens does not prove a causal relationship with this abortion event. In addition, serology might be negative in the case of sampling at or shortly after an abortion since seroconversion takes time.

In our study, we combined an easy sampling method with both culture and PCR. With culturing, both on selective and non-selective media, we cover a wide range of bacterial pathogens in potential. This was standard procedure in our laboratory. However, since most cases of abortions are caused by a limited number of bacteria species, culture can be replaced by some species-specific PCRs. Such PCRs are widely used for abortion diagnostics in small ruminants [15], and their use in combination with our easy sampling method seems highly promising. The fact that one DNA-isolation step can be used for the different PCR assays lowers the costs per test. This approach has advantages for known causes of abortion. A disadvantage is that it is impossible to perform PCRs for unknown agents. However, as we have shown, samples collected with both our alternative methods, the oropharynx mucus samples, and the fetal lung puncture samples, can be used for performing a PCR. Therefore, a multiplex PCR can easily be broadened to other agents of interest for a specific region. 

The step-wise investigative approach described by Borel et al. (2014) [6] is the most proper way to investigate small ruminant abortions. The alternatives we investigated are not advised as a replacement for post-mortem abortion diagnostics but as an additional, easily accessible low-threshold tool for practitioners and farmers to provide the first indication when confronted with abortion problems. 

## 4. Materials and Methods

### 4.1. Post-Mortem Examination

A post-mortem examination was performed according to a standardized protocol that was described before [5,7]. A submission always contained one or more fetuses. A submission was considered complete when both fetus and placental membranes were included. 

### 4.2. Sampling from Oropharynx and Lung

In addition to the normal protocol, sampling the fetus was done in two alternative ways to mimic how the owner or veterinarian might sample the fetuses. Mucus deep from the oropharynx was sampled from each submitted fetus with an Eswab (Copan Diagnostics Inc., Murrieta, CA, USA) by the oral route and stored in an Eswab medium. Additionally, lung puncture samples were collected by per thoracal fine needle aspiration (18G- needle, Microlance) directly behind the scapula, halfway the length of the scapula (Figure 1), and this collected material was also stored in Eswab medium. Sampling was done after identification of the submission and before regular post-mortem examination commenced.

### 4.3. Bacteriology and PCR

#### 4.3.1. Bacteriological Culture

The two alternative samples were used for bacteriological culture, which was performed conforming to standard procedures that were also applied for normal pathological samples. For this purpose, XLD-agar plates were inoculated to detect *Salmonella* spp, sheep blood agar (SBA) plates for the detection of *Listeria* spp., *E. coli*, *Yersinia* spp. or *T. pyogenes*, and CCDA and Skirrov agar plates for the detection of *Campylobacter* spp. All plates were incubated at 37 ± 1 °C. XLD and SBA plates were incubated aerobically, and CCDA and Skirrow plates at microaerophilic conditions. XLD plates were checked for growth after one day of culturing, SBA after one and two days, and CCDA and Skirrow after two and three days. Suspected colonies were identified using a MALDI-TOF system with standard software supplies by the manufacturer. 

#### 4.3.2. PCR for *Chlamydia* spp. and *Coxiella burnetii*

In addition to bacteriological culture, the two alternative samples were tested with a commercial PCR for DNA of *C. burnetii* and *Chlamydia* spp. 

DNA was extracted using an AM1840 MagMAX™ Total Nucleic Acid Isolation Kit (Thermo Fisher Scientific, Waltham, MA, USA) according to the manufacturer’s instructions. For detection of *Chlamydia* spp., and distinction of *C. psittaci* and *C. abortus* from other *Chlamydia* spp., the following real-time PCR was used: 0.3 µM universal forward primer 5′-CCTTAAGTCGTTGACTCA ACC-3′, 0.3 µM universal reverse primer 5′-AAA YRCTTG CCCAACCTA GTC-3′, 0.1 µM *Chlamydia* spp. universal probe YY–CGCCCAAGGTGAGGCTGATGA–BHQ1, and 0.075 µM *C. psittaci/C. abortus* specific probe FAM–AACCGTCCTAAGACAGTTATCCTTATCCTT–BHQ1. The real-time PCR was performed on an ABI7500 fast thermal cycler (Thermo Fisher Scientific, Waltham, MA, USA) using TaqMan universal master mix (Thermo Fisher Scientific) under the following conditions: denaturation 15 min at 95 °C, followed by 40 cycles with 15 s at 95 °C, 60 s at 55 °C, and 60 s at 72 °C. Data were analyzed (with ROX) using a Delta Rn threshold of 0.2 for the FAM signal and 0.1 for the YY signal. PCR cycles 3 to 15 were used for baseline normalization. Samples with a Ct <40 for *Chlamydia* spp., and a Ct <40 for *C. psittaci* and *C. abortus* were regarded positive for *C. psittaci* or *C. abortus*. Samples with a Ct <40 for *Chlamydia* spp., and a Ct >40 or no signal for *C. psittaci* and *C. abortus* were regarded *Chlamydia* spp. positive not being *C. psittaci* or *C. abortus*. Samples without a signal for *Chlamydia* spp., *C. psittaci* and *C. abortus* were regarded negative for *Chlamydia* spp.

Samples positive in the *Chlamydia* spp. PCR but negative in the *C. psittaci* and *C. abortus* PCR were subsequently tested in the *C. pecorum* KASP PCR using the following primers; forward primer Cpecorum_CPC_4_Specific 5′-GAAGGTCGGAGTCAACGGATTGTCCACATGAGTCAAAAATTACTTGG-3′ and reverse primer Cpecorum_CPC_4_Common 5′-AATCACAAGCGTAAGAAGCGAA-3′. The real-time PCR was performed on an ABI7500 fast thermal cycler (Thermo Fisher Scientific) using the KaspRT reagent (LGC Genomics, Hoddesdon, UK) and the following steps: initial denaturation for 15 min at 95 °C followed by 45 cycles of denaturation for 20 seconds at 95 °C, and annealing and extension for 60 s at 57 °C (fast modus). Data were analyzed (with ROX) using a Delta Rn threshold of 0.2 for the FAM signal. PCR cycles 3 to 15 were used for baseline normalization. Samples with a Ct < 40 were regarded positive for *C. pecorum*.

*C. burnetii* was detected using the TaqVet *Coxiella burnetii* Absolute Quantification kit (LSI, Lissieu, France) according to the instructions in the manual. A sample was called *C. burnetii* positive when more than 10^3^ bacteria/mL were detected, using the calibration series of the manufacturer [23].

### 4.4. Data Analysis

Results of pathological examination were presented as a series of findings resulting in a final diagnosis. For analytical purposes, only the final diagnosis remained. The pathology results were used as a reference to which the results of the oropharynx and fetal lung samples were compared. For all three sampling methods, all *Chlamydia *spp. and *Campylobacter* spp. were aggregated at the genus level.

Descriptive statistics were used for most of the comparisons of the test results. In addition, the sensitivity (Se) of each of the tests conducted on the alternative matrices was calculated by dividing the number of oropharynx or lung samples where bacteria were correctly detected by the number of pathology results where the same bacteria were cultured (Se = a/(a + c)) [24].

The specificity (Sp) was calculated by dividing the number of samples that were correctly classified as test negative in the oropharynx or lung sample by the number of negative pathology results (Sp = d/(b + d)). It was assumed that the first isolation from a fetus or submission was the only one. The sensitivity and specificity are presented as percentages with the accompanying 95% confidence intervals. 

The agreement between the pathology results and an alternative testing strategy results was presented, using Cohen’s Kappa measure for agreement. We divided the actual agreement beyond chance by the potential agreement beyond chance. More detailed information on the calculation method can be found in Cohen (1960) [25].

## 5. Conclusions

In conclusion, although a full step-wise post-mortem examination remains the most proper way of investigating small ruminant abortions, the easily accessible, low-threshold tool for practitioners and farmers as described in this paper not only provides reliable results compared to results of the post-mortem examination but also stimulates farmers and veterinarians to submit fetuses and placentas if necessary. Suggestions for further improvement of both alternatives have been summarized. Both alternatives could also be tailor-made for specific regions with their specific causes of abortion. 

## Figures and Tables

**Figure 1 pathogens-10-01147-f001:**
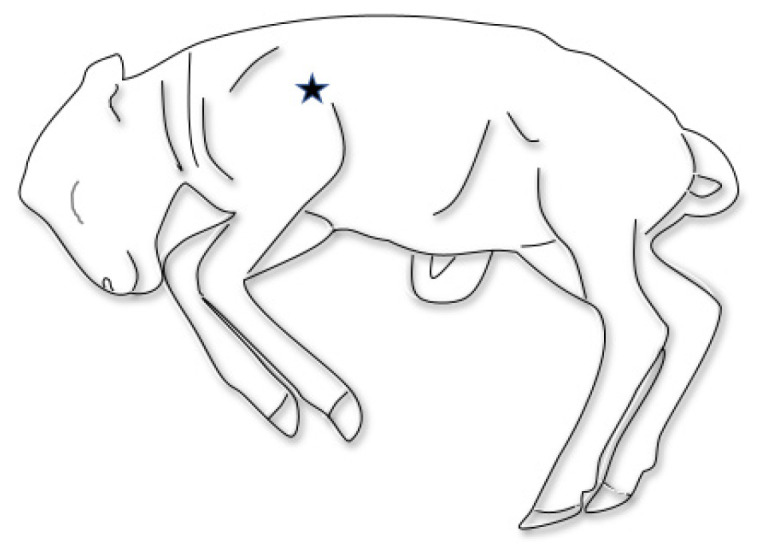
Location of sampling of the fetal lung.

**Table 1 pathogens-10-01147-t001:** Sampling strategy with descriptive numbers of submissions, unique farms and fetuses, and their origin.

	Submissions	Unique Farms	Fetuses
Ovine	48	45	83
Caprine	28	23	63
Total	76	68	146

**Table 2 pathogens-10-01147-t002:** Results of fetal oropharynx sample versus results of pathological examination for the main bacterial causes of abortion in the Netherlands in 2015–2016.

	Pathology				
Oropharynx Sample	*Chlamydia* spp.	*Campylobacter* spp.	*Listeria* spp.	Negative	Total
*Chlamydia* spp.	15	0	0	11	26
*Campylobacter* spp.	0	6	0	4	10
*Listeria* spp.	0	0	6	0	6
Negative	2	2		99	103
Total	17	8	6	114	145

**Table 3 pathogens-10-01147-t003:** Results of fetal lung sample versus results of pathological examination for the main bacterial causes of abortion in the Netherlands in 2015–2016.

	Pathology				
Lung Sample	*Chlamydia* spp.	*Campylobacter* spp.	*Listeria* spp.	Negative	Total
*Chlamydia* spp.	16	0	0	13	29
*Campylobacter* spp.	0	6	0	0	6
*Listeria* spp.	0	0	7	0	7
Negative	1	2	0	101	104
Total	17	8	7	114	146

**Table 4 pathogens-10-01147-t004:** Sensitivity and specificity of the fetal oropharynx sampling in comparison with pathological examination.

	Sensitivity	95% CI	Specificity	95% CI
*Chlamydia* spp.	88.2	82.9–100	89.9	84.7–95.1
*Listeria* spp.	100	100–100	100	100–100
*Campylobacter* spp.	75	45–100	100	100–100
*Coxiella burnetii*	ntd	ntd	100	100–100
Total	90.6	80.5–100	97.6	96.3–98.9

**Table 5 pathogens-10-01147-t005:** Sensitivity and specificity of fetal lung sampling in comparison with pathological examination.

	Sensitivity	95% CI	Specificity	95% CI
*Chlamydia* spp.	94.1	72.9–100	91.4	86.6–96.3
*Listeria* spp.	100	100–100	100	100–100
*Campylobacter* spp.	75	45–100	97.1	94.3–99.9
*Coxiella burnetii*	ntd	ntd	97.9	95.5–100
Total	87.1	75.3–98.9	97.3	95.8–98.9

CI: confidence interval; ntd: not to determine.

## Data Availability

An overview of results of pathological examination as presented in this paper is published in the small ruminant half-yearly monitoring and surveillance report of the first half of 2016 (Rapportage Monitoring Diergezondheid Kleine Herkauwers. Eerste halfjaar 2016). The remaining data presented in this study are available on request from the corresponding author. These data are not publicly available due to privacy reasons.

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
