# Peer review of "An Accessible Diagnostic Toolbox to Detect Bacterial Causes of Ovine and Caprine Abortion"

_pathogens, 2021, doi:10.3390/pathogens10091147_

Round 1

Reviewer 1 Report

The manuscript by Renévan den Brom and colleagues describes the analysis of two alternative sampling systems for the investigation of bacterial causes of small ruminants abortions. The manuscript is clearly written and logically organized. The content is technically sound, and overall, the research is well described. The conclusions are supported by the analysis of the data presented. 

I really enjoyed reading this article, and I have only a few minor revisions:

-Not all bacterial agents names are italicized. Also, after writing the complete name in the first mention, the genus name can be shortened to just the capital letter. Please check and correct (i.e. lines 121 to 130; line 139; ).

-Some sentences are too long. It would make it easier if shorter sentences were made. As an example: lines 32-36; 109-112; ...

-Add "(Table 4)" in lines 173 ("...was 97.3%.") and 179 ("...was 87.1%.").

-Table 1: in the first column, please replace "lung sample" by "oropharynx sample". I believe that was a mistake.

-Table 3 and Table 4: please correct the sensitivity values for Chlamydia spp. in table 3 (fetal orophanynx samples) should be 88.2 % (15/17); in table 4 (fetal lung samples) should be 94.1% (16/17).

Reviewer 2 Report

This paper explored the feasibility of two alternative sampling systems for the investigation of bacterial causes of ovine and caprine abortions. Test results of these samples were compared with each other, and with results from standard post-mortem examinations.

Minor change of english language spell is required

Line 27 Delete own (redundant)

Line 91 and 106  puncture

Lines 120-130 Use italics for germ names

Lines 204-205, 258-259 Check spaces and spelling

Please follow the template

The references do not follow the guidelines (numbers in the text, doi, italics).
